# Association of Emotion Regulation and Dispositional Mindfulness in an Adolescent Sample: The Mediational Role of Time Perspective

**DOI:** 10.3390/children10010024

**Published:** 2022-12-23

**Authors:** Cristián Oyanadel, Yalín Núñez, Melissa González-Loyola, Ignacio Jofré, Wenceslao Peñate

**Affiliations:** 1Department of Psychology, Universidad de Concepción, Concepción 4030000, Chile; 2Department of Clinical Psychology, Psychobiology and Methodology, Universidad de La Laguna, 38200 La Laguna, Spain

**Keywords:** time perspective, mindfulness, emotional regulation, adolescents

## Abstract

This study relates emotional regulation strategies with dispositional mindfulness and the mediating role of time perspective. It is based on the fact that one of the mechanisms of mindfulness consists in providing protective emotional regulation strategies. At the same time, a direct relationship between dispositional mindfulness and time perspective has been observed. To do this, a representative sample of 320 Chilean adolescents from the city of Talcahuano, whose age ranged between 14 and 17 years old, and who were attending high school, was evaluated. The Zimbardo Time Perspective Inventory, the Difficulties in Emotion Regulation Scale, and the Five Facet Mindfulness Questionnaire were applied. Regression analysis results verified the close relationship between emotional regulation and dispositional mindfulness (R^2^ = 0.54), as well as with the factors of time perspective (R^2^ = 0.41), explaining, between both of them, 60% of the variance of difficulties in emotional regulation. The possible mediational role of time perspective between dispositional mindfulness and emotional regulation is established.

## 1. Introduction

Adolescence is considered as one of the most important transition stages in a human being’s life, characterized by an accelerated growth rate and accompanied by great physical and psychological changes [1]. These changes transform this stage into a period associated with the presence of risk behaviors, and high vulnerability for the appearance of mental health problems [2,3,4]. In this sense, emotional regulation processes may be a key element in this vulnerability, and the main related concepts will be examined.

Emotion regulation is defined as “the process of initiating, maintaining, modulating, or changing the occurrence, intensity, or duration of internal affective states and physiological processes, often with the aim of achieving a goal” [5] and influences the way in which adolescents can experience these transformations corresponding to their stage of development [6]. In adolescence, this ability involves managing and organizing internal components (neurobiological, cognitive, and subjective evaluations), behavioral components (facial and behavioral actions), and social components (values, social context, personal goals) [7].

However, when emotional regulation presents difficulties, it becomes ineffective and maladaptive and can lead to a loss of the functional value of emotions [8]. Available data show how difficulties in regulating emotions during childhood and adolescence are associated with a large portion of mental disorders, including personality disorders [3,8,9], depressive disorders and anxious symptoms [3]. It has even been observed that distress expression behaviors are associated with child distress regulation behaviors (such as disengagement of attention, parent-focused behaviors, and self-soothing) in babies from one to three years old [10].

On the other hand, adequate emotional regulation skills constitute a protective factor for mental health [8]. In this sense, it has been observed that dispositional mindfulness is directly related to these emotional regulation skills [11,12,13,14,15,16].

The conceptualization of mindfulness, as a dispositional trait, is related to a set of characteristics or traits present in the daily life of individuals. These include, among others, the ability to not act with reactivity towards internal experiences, and the ability to observe, pay attention and adequately deal with sensations, perceptions, thoughts and feelings, act with awareness and not judge the experience [12,17,18]. In line with this, the presence of dispositional mindfulness is considered a protective factor against the appearance of mental health disorders [14,15,19,20].

A third variable is time perspective. Available data support the existence of an association between time perspective and a high degree of dispositional mindfulness [20]. The concept of time perspective [21] refers to people’s attitudes towards past, present and future time zones. The established temporal dimensions are the positive past, negative past, hedonistic present, fatalistic present, future (goal-oriented) and transcendental future beyond death [22].

Regarding adolescence, it has been observed that time perspective could be related to the process of identity formation, the ability to adapt to the environment and the capacity for self-regulation [23]. Available data indicate that adolescents are more focused on a hedonistic present [24] and less on the future [25]. Different studies have observed that there is no ideal time dimension, but rather the existence of a balanced profile that would be determined by specific indexes of each orientation (past, present and future). Basically, a protective profile would be related to a balanced profile in relation to a positive past, a hedonistic present at medium levels, and an orientation towards the future supported by goals—this profile being associated with a higher level of health and well-being [20,26,27]. On the other hand, an unbalanced profile has been related to greater discomfort, even having a vicarious effect, in which an unbalanced perspective of parents has been related to their children’s health [28].

In relation to emotional regulation, studies have found a relationship between a greater capacity for self-regulation and a more positive time orientation [29], as it has also been observed that students with high levels of self-regulation have lower levels of negative past [23,30]. Thus, emotional regulation has been observed to be related to both time perspective and mindfulness.

Based on this background, this research aimed to determine whether the relationship between emotional regulation and dispositional mindfulness can be associated with adolescents’ time perspectives.

## 2. Method

This research is a retrospective ex post facto design of a group [31]. Difficulties in emotional regulation have been taken as a predictor variable, and the five facets of mindfulness and the five factors of time perspective have been used as predictors. Additionally, the role of age, sex, socioeconomic level and whether or not they are receiving psychotherapeutic help in relation to emotional regulation have been taken into account.

## 3. Participants

A multi-stage sampling of municipal and private subsidized schools was carried out, with a total of 8 schools (3 municipal and 5 private subsidized), including 320 adolescents (140 men [44%] and 180 women [56%]), attending secondary school, with an average age of 15 years old (14 years old [27%], 15 years old [26%], 16 years old [30%], 17 years old [17%]; s.d. = 1.03). The only inclusion criterion was to be a high school student, and recruitment was by convenience and on a voluntary basis. Participants and his/her tutor signed an Informed Consent agreement. This study was approved by the Ethics and Bioethics Committee of the Department of Psychology, Universidad de Concepción (ref. 2008–2018).

## 4. Instruments

Sociodemographic survey: through a survey developed for this study, information was collected on age, sex, socioeconomic level, average grades obtained in the last semester and participation in mental health treatment.Scale of Difficulties in Emotional Regulation (DERS): Chilean adapted version by Guzmán et al., (2014) [32]. It is a Likert scale, scored between 1 and 5, composed of 36 items designed to evaluate the difficulties in emotional regulation in boys and girls. For this study, an adaptation was made for adolescents, through modifications in the language after cognitive interviews and evaluation of the judgment of 6 experts. The scale consists of 5 subscales, which allow for obtaining a general score, including aspects such as: (1) emotional rejection, related to the ability to accept one’s own discomfort; (2) emotional lack of control, referring to the difficulties in maintaining control when negative emotions are experienced; (3) daily interference, referring to difficulties in concentrating and completing tasks while experiencing negative emotions; (4) emotional neglect, difficulties to attend and have knowledge of emotions; (5) emotional confusion, related to the difficulty to identify and be clear about the emotions that are experienced. In the Chilean validation [32], levels of internal consistency (Cronbach’s α) ranged from 0.66 to 0.89 were obtained, whereas the internal consistency obtained in the sample of this study ranged from 0.69 to 0.92.Five Facet Mindfulness Questionnaire (FFMQ): Chilean adapted version by Schmidt and Vinet (2015). The FFMQ consists of 39 items distributed in 5 factors. It is answered through a Likert scale with a range from 1 (never or very rarely true) to 5 (very often or always true), with a minimum score of 39 points and a maximum of 195 points. It was developed to assess the tendency of people to act attentively considering 5 factors: observing, describing, acting with awareness, not judging internal experiences and not reacting to internal experiences. These five facets also allow obtaining an overall score in dispositional mindfulness. The instrument was developed and validated to its application in adults from the age of 18, so for this study, small changes were made to adapt it for a sample of adolescents by means of cognitive interviews and expert assessment. The adults adapted version [18] obtained an α ranging from 0.62 to 0.86, and for the present study, internal consistencies ranged from 0.63 to 0.81.Zimbardo Time Orientation Inventory (ZTPI, [21]): Chilean version by Oyanadel, Buela-Casal & Perez-Fortis (2014) [33]. It is an inventory composed of 56 items, answered through a Likert scale ranging from 1 to 5 (totally disagree to totally agree). It evaluates the temporal orientation in its 5 facets: negative past, positive past, hedonistic present, fatalistic present and future. In turn, these primary factors allow the construction of two complex factors: the deviation from the balanced time perspective (DBTP), a measure in which the higher the score, the greater the distance from a balanced adaptive profile; and the deviation from the negative time perspective (DNTP), where the higher the score, the further away from a negative time profile. In this research the Chilean version [33] had a Cronbach’s α ranging from 0.70 to 0.81.

## 5. Procedure

Prior to the application of the instruments, cognitive adaptation stages were carried out for the scales used. This was carried out in order to minimize possible biases in the application of the scales to the adolescent population. These previous stages were cognitive interviews and an expert panel.

### 5.1. Cognitive Interviews

Cognitive interviews were conducted in adolescents, whose ages ranged from 14 to 17 years old (N = 5), from municipal and private subsidized schools. Participants were instructed to read the DERS-E and FFMQ aloud and ask any questions they could have about instructions, questions and response options. In this procedure, each response item was discussed with the participants at this stage to assess their understanding [34,35,36]. In this regard, there were substitutions of some words more commonly used in the Spanish spoken in Chile, but not in the sense of the items. 

### 5.2. Experts’ Panel

A panel of six experts from the fields of psychology, psychiatry and education, with high knowledge on mindfulness, emotions and adolescence, analyzed the instruments and suggested further modifications according to: (a) Comprehension: an adolescent would understand the logic of the question; (b) Context: words being used correspond to adolescents’ ways of expressing themselves; (c) Precision: there are no potential problems for adolescents to remember the information that is requested; (d) Respect: words used in the test are respectful and do not hurt feelings. In this regard, the expert judges unanimously confirmed all the changes suggested at the previous stage. In this sense, formal changes were conducted for the DERS-E scale (ítems 4, 5, 8, 10, 14, 24 y 25), and FFMQ (ítems 1, 2, 5, 12, 19, 23, 28, 32, 36, 37).

Once inventory contents were clarified, the next step was the sample recruitment. The selection of classes was carried out randomly, without the intervention of the research team; however, in some schools there was only one class per level, therefore it was not possible to make a random selection of classes. Instruments were applied by the second author, together with fifth-year psychology students previously trained in the application of all the instruments. Students were given 45 min to answer the instruments, during hours selected by the school administration. Previously, the informed consent form had been signed by parents/guardians.

## 6. Data Analysis

ANOVAs were carried out to contrast the role of the sociodemographic factors (sex, age, socioeconomic level and whether or not mental health treatment was being received), and difficulties in emotional regulation (total DERS score). Sex was divided into male and female; age was divided into four groups (14, 15, 16 and 17); socioeconomic level was also divided into four groups (upper middle income, average income, lower middle income and low income); and receiving mental health treatment was divided into two groups (yes and no).

To analyze the possible predictive power on these regulation difficulties and the variables of dispositional mindfulness and time perspective, multiple regression analyses (stepwise method and hierarchical) were carried out. Two regression analyses were separately performed for the predictive role of dispositional mindfulness and time perspective. A third analysis was carried out taking together both groups of variables. Finally, to determine the mediational role of time perspective, between dispositional mindfulness and difficulties in emotional regulation, a mediational analysis was carried out using the PROCESS software for SPSS [37]. In this case, only general scores were used. For time perspective, deviation from the balanced time profile was used as the general score. DBTP was obtained according to the following formula [27]:DBTP = **√** (oPN − ePN)^2^ + (oPP − ePP)^2^ + (oPF − ePF)^2^ + (oPH − ePH)^2^ + (oF − eF)^2^
where oPN is the optimal score for PN (Past Negative) and ePN is the empirical or observed score. The optimal values for all dimensions are oPN = 1.95; oPP (Past Positive) = 4.60; oPF (Present Fatalism) = 1.50; oPH (Present Hedonism) = 3.90; oF (Future) = 4.00.

## 7. Results

As a preliminary analysis, ANOVAs were carried out, together with the total score on the DERS scale to study if there were any differences between sociodemographic and clinical variables (sex, age, socioeconomic level and whether or not they receive psychotherapy), and levels of emotional regulation.

No significant differences were found according to the socioeconomic level of the participants (F(0.340) = 2.2), nor in relation to their age (F(3.340) = 1.17). However, significant differences were observed in difficulties in emotional regulation by gender (F(1.342) = 10.42; p = 0.001), with greater difficulties in women, and if they received psychotherapy or not (F(1.341) = 13.0; p = 0.000), with greater difficulties for those receiving psychotherapy.

Regarding the predictive analysis of the dispositional mindfulness variables and the time perspective variables on emotional regulation difficulties, multiple regression analyses (stepwise method) were first carried out separately for each group of variables (Table 1). Both analyses provided a significant prediction model: for the dispositional mindfulness variables, the model obtained an F = 103.47 (p = 0.000) with a final R^2^ of 0.54; and in the case of time perspective factors, an F = 59.68 (p = 0.000) was obtained with a final explained variance of R^2^ = 0.41.

In the case of dispositional mindfulness, four of the five facets entered the equation (the observe facet’s participation was not significant), all with a negative weight (greater difficulties in emotional regulation with lower levels of dispositional mindfulness). As for time perspective, four factors also entered the equation. The greatest significant weight corresponded to negative past, and only the future perspective was negatively associated (the lower the levels of goal-oriented future perspective, the greater the difficulties in emotional regulation).

With these data in mind, a new hierarchical multiple regression analysis was carried out, first introducing the dispositional mindfulness variables, secondly the time perspective variables, and thirdly the two variables that in the mean contrasts offered significant results (gender and whether or not they received mental health treatment (n = 82, 26% outpatient). The final model (F = 86.42; p = 0.000) had higher explained variance than the previous two models (R^2^ = 0.6) (Table 2).

As it can be seen out of the twelve possible variables, six of them entered the equation. Four of them belong to dispositional mindfulness (observe facet being left out), and one to time perspective (past negative), which had the greatest weight in the equation. Sex was also part of the equation, but in fifth place, and the variable related to receiving mental health treatment did not reach significance.

With these data in mind, and in order to analyze the possible mediational role of time perspective variables on the relationship between emotional regulation difficulties and dispositional mindfulness, mediational analyses were carried out. For procedural reasons, the total scores on each of the parameters were used (Figure 1).

As can be observed, there is a direct relationship between dispositional mindfulness and difficulties in emotional regulation (total effect (c) = −0.895, p = 0.001). This relationship is diminished in a statistically significant way by including time perspective as a mediating variable in this relationship, with dispositional mindfulness having a direct effect smaller than the total effect (c′ = −0.700, p = 0.001), and having an indirect effect through the mediating role of DBTP on difficulties in emotional regulation (ab = −0.195, p = 0.001, 95% CI, from −0.273 to −0.126).

## 8. Discussion

The aim of this investigation was to know the relationship between the emotional regulation capacity of adolescents, their temporal profile and dispositional mindfulness. The results confirm the close relationship between emotional regulation and dispositional mindfulness. Likewise, emotional regulation was closely related to time perspective.

Specifically, the participation of the facets of dispositional mindfulness corresponded to four out of five (where the first two, acting with awareness and not judging internal experiences, already explained 48% of the variance of the final 54%). These data confirm how adolescents with fewer difficulties in emotional regulation have higher levels of dispositional mindfulness; these data coincide with neuroscience research in relation to the association between mindfulness practices and improvements in emotional regulation processes [38,39].

Data show that adolescents who present a temporal profile based on a time perspective of negative past, hedonistic and fatalistic present, and do not have a future perspective based on goals and purposes, are the ones who show greater adjustment difficulties in emotional regulation. It is remarkable that the perspective of the positive past does not participate in the prediction. On the other hand, the relationship between the hedonistic present and regulation difficulties may be due to the peculiarity of this temporal perspective, since both low levels of this variable and high levels are associated with worse emotional conditions [27]. Previous research has already linked the balanced temporal profile with better health and well-being conditions, including better self-regulation capacities [30]. This also extends to the observed direct relationship between negative past perspective and emotional dysregulation.

The greater emotional regulation difficulties of those adolescents who are receiving therapeutic support can be understood as supporting the internal validity of the research, since it is expected that people who request psychological help suffer from a lack of adequate emotional regulation coping strategies [40].

With regard to the joint participation of the facets of dispositional mindfulness and time perspective in the prediction of emotional dysregulation, the four facets of mindfulness mentioned above participate again, but only one, the negative past, does in the case of time perspective. However, this variable is the first to enter the equation and explains 36% of the variance, of the total 60%. This participation favors the mediational role of time perspective in the relationship between dispositional mindfulness and emotional regulation. Specifically, a balanced time perspective strengthens the relationship between protective emotional regulation strategies and mindfulness, and a negative past time perspective may be influencing the limitation of mindfulness to favor adaptive emotional regulation strategies [13].

It is interesting that DBTP is a variable that affects the relationship between mindfulness and emotional dysregulation. In this sense, moving away from a balanced time perspective can decrease the ability of mindfulness to regulate emotions, which is one of its most characteristic aspects. The mediational role of DBTP has already been studied, and it has been shown that this variable is strongly related to psychological flexibility [41]. In this way, the deviant temporal profile decreases the ability of dispositional mindfulness to regulate emotions and modify maladaptive regulation patterns. If we consider that the disposition to mindfulness can be a variable that favors emotional regulation in adolescence, suffering from changes in balanced time perspective can lead to significant difficulties for development [42].

Certainly, this assertion needs further empirical support, with research designs that allow for the establishment of this sequence of relationships. It would be interesting to conduct longitudinal studies to test this relationship between time perspective, dispositional mindfulness and emotional dysregulation with instruments specially developed for the adolescent population [43].

Regarding the limitations of the study, first of all, this is an observational, correlational study, so cohort or experimental designs would be needed to clearly establish causal and mediational relationships. In addition, although it is a representative sample, it is a limited sample in its size and scope, since it is only representative of a region of Chile, not of the entire population. On the other hand, there is no clinical sample where emotional regulation strategies are linked to suffering from a mental disorder. It is true that part of the sample stated that they were receiving psychological support, but there is no accreditation of the type of support received or the disorder or problem being treated.

As for future proposals, it is considered pertinent to study the impact of interventions that favor the capacity for emotional regulation, through training in balanced temporal profiles and mindfulness strategies. Previous studies have already generated evidence that it is possible to change the profile of time orientation, towards more “balanced” levels, through interventions that enhance orientation to the future and the positive past, reduce the influence of the negative past and fatalistic present, and regulate the hedonistic present; with this, we could support adolescents to have a better capacity for emotional regulation, better mental health, and a stronger process of their identity development [44,45]. In addition to this, mindfulness training would also contribute to stability in emotional functioning, and there is already evidence to consider it a protective factor against the appearance of mental health disorders [14,15,19,20,22].

To summarize, studies to date have emphasized the relationship between temporality and emotional regulation, or between dispositional mindfulness and emotional regulation, in adults or emerging adults. Research on the adolescent population is scarce; therefore, this study allows us to affirm that there is a significant relationship between time profiles, dispositional mindfulness and emotional regulation, so that adolescents with a balanced time profile have fewer emotional regulation difficulties and higher levels of dispositional mindfulness.

## 9. Conclusions

The current study confirmed the close relationship between emotional regulation with dispositional mindfulness and time perspective. Results suggest that acting with awareness and not judging internal experiences are especially relevant for emotional regulation. Additionally, adolescents who focus on negative past, hedonistic and fatalistic present, and do not have a future perspective based on goals and purposes, are typically the ones with greater adjustment difficulties in emotional regulation. On the other hand, a balanced time perspective would strengthen the relationship between protective emotional regulation strategies and mindfulness. Based on these results, practitioners and healthcare providers in schools should consider implementing interventions that favor the development of balanced time perspectives and mindfulness strategies.

## Figures and Tables

**Figure 1 children-10-00024-f001:**
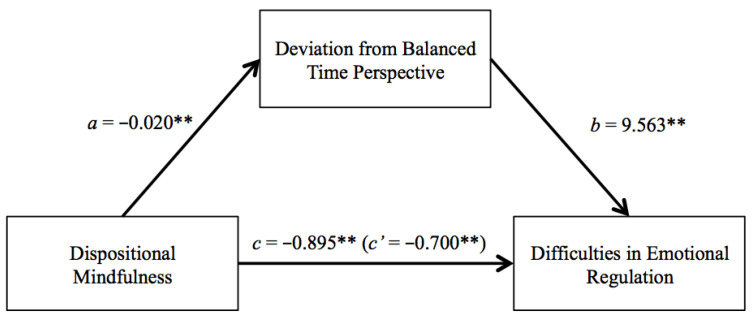
Mediation of time perspective in the relationship between the level of dispositional mindfulness and emotional regulation difficulties. **: p < 0.001.

**Table 1 children-10-00024-t001:** Multiple Regression Analyses with Mindfulness and Time perspective variables (step-by-step method) in predicting general level of difficulties in emotional regulation (N = 320).

				95% C.I.	
	B	t	p	Inferior	Superior
Dispositional mindfulness variables					
(Constant)		27.68	0.000	150.29	173.28
Acting with awareness	−0.35	−8.30	0.000	−1.39	−0.86
Not judging internal experiences	−0.46	−11.03	0.000	−1.74	−1.21
Describing	−0.19	−4.17	0.000	−0.96	−0.35
Not reacting to internal experiences	−0.16	−3.80	0.000	−0.91	−0.20
Time perspective variables					
(Constant)		0.11	0.917	−17.35	19.30
Negative past	0.53	11.96	0.000	14.21	19.80
Future	−0.14	−3.32	0.001	−8.27	−2.11
Hedonist present	0.12	2.77	0.006	1.41	8.29
Fatalistic present	0.09	2.02	0.044	0.12	8.87

Note: B = Standardized beta coefficient; t = t-test; p = probability; C.I. = Confidence Interval.

**Table 2 children-10-00024-t002:** Hierarchical Multiple Regression Analysis predicting difficulties in emotional regulation, taking together both time perspective variables and dispositional mindfulness variables (N = 320).

				95% C.I.	
	B	t	p	Inferior	Superior
(Constant)		10.77	0.000	85.5	123.69
Negative past	0.27	6.26	0.000	5.85	11.20
Acting with awareness	−0.29	−6.99	0.000	−1.17	−0.66
Not judging internal experiences	−0.34	−7.85	0.000	−1.36	−0.81
Not reacting to internal experiences	−0.13	−3.23	0.001	−0.77	−0.19
Sex	0.13	3.87	0.000	2.75	8.45
Describing	−0.14	−3.37	0.001	−0.79	−0.21

Note: B = Standardized beta coefficient; t = t-test; p = probability; C.I. = Confidence Interval.

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
