# Peer review of "Association of Emotion Regulation and Dispositional Mindfulness in an Adolescent Sample: The Mediational Role of Time Perspective"

_children, 2022, doi:10.3390/children10010024_

Round 1

Reviewer 1 Report

In their manuscript "Association of Emotion Regulation and Dispositional Mindful- 2 ness in an Infant-Adolescent Sample: The Mediational Role of 3 Time Perspective" Oyanadel-Véliz et al. describe the close relationship between emotional regulation with dispositional mindfulness and time  perspective. 

Overall the manuscript is easy to read, the introduction section provides a clear and stringent background, the methods and results section is presented clear and the discussion is concise.

Minor points to concider:

1) Page 3, line 85: please define "use of pharmacotherapy". Which informations in detail were available? Actual therapy or lifetime therapy? Psychopharmacotherapy or any pharmacotherapy?

2) Page 3 line 87 - please delete one "to"

Author Response

Dear Reviewer:

Thank you for your kind review, and pointing out our mistakes

- P.3, line 85, the type of question in the questionnaire is clarified, and line 88 one “to” is erased.

Thank you for your time and consideration,

Best regards,

Wenceslao Peñate, PhD

Universidad de La Laguna

Spain

Reviewer 2 Report

Dear  Authors,

I think that your study is original but I have some suggestions. You find my suggestions in the pdf file.

Best regards

Author Response

Dear Reviewer:

Thank you for your review. Their appreciations have allowed us to improve our manuscript.

- Regarding the use of the term child-adolescent, since there is no clear cut-off regarding the ages in these periods, we prefer to use this term, classic in psychopathology, to differentiate it from the young and adult stages. But we understand that if the term 'adolescent' is more comprehensive, we have proceeded to change it throughout the manuscript.

- Improvements are introduced in the introduction and the central objective of this work is clarified.

- Aspects related to participants are clarified as requested, and considering the type of design.

- Your contribution has been very significant in clarifying the methodological and analytical steps, which have been updated in the manuscript.

- The references in the discussion have been updated.

Thank you for your time and consideration,

Best regards,

Wenceslao Peñate, PhD

Universidad de La Laguna

Spain

Round 2

Reviewer 2 Report

Dear Authors,

Unfortunately, I see that the revisions were not done carefully. The contribution of contemporary literature to the discussion should have been increased, but these authors did not properly edit it. At the same time, the statistical method should be explained in more detail.

Best regards

Author Response

Dear Reviewer 2:

Enclosed you will find our revised manuscript, entitled: “Association of Emotion Regulation and Dispositional Mindfulness in an Adolescent Sample: The Mediational Role of Time Perspective”. We sincerely appreciate the revision process, and the suggestions you made. We have taken all of your suggestions into account, and made the respective changes, in order to improve the article’s quality.

Some of the changes include:

- A specific subsection on data analysis has been created, which has reorganized and expanded the information on statistical procedures. To better specify, the titles of the tables have been modified.

- The implications of the results have been discussed, which are interesting in light of current research. The introduction has also been supplemented.

Thank you for your time and consideration,

Best regards,

Wenceslao Peñate, PhD

Universidad de La Laguna

Spain